# Prevalence and Predictors of Self-Medication Among Ophthalmic Patients in Jordan: A Cross-Sectional Analysis

**DOI:** 10.3390/healthcare13040372

**Published:** 2025-02-10

**Authors:** Alyaa Ismael Ahmad, Mohammad Akram Abdeljalil Huwari, Alaa A. Alsharif, Hamzeh Mohammad Alrawashdeh, Abdallah Y. Naser

**Affiliations:** 1Department of Applied Pharmaceutical Sciences and Clinical Pharmacy, Faculty of Pharmacy, Isra University, Amman P.O. Box 11622, Jordan; alyaa.ahmad00@gmail.com; 2Department of Ophthalmology, Jordan Eye Center, Amman 11954, Jordan; m_huwari@hotmail.com; 3Department of Pharmacy Practice, College of Pharmacy, Princess Nourah bint Abdulrahman University, P.O. Box 84428, Riyadh 11671, Saudi Arabia; aaalsharif@pnu.edu.sa; 4Department of Ophthalmology, National University Hospital, Singapore 119228, Singapore; dr_hmsr@yahoo.com

**Keywords:** eye disease, Jordan, ophthalmology, self-medication

## Abstract

**Background:** When people decide to treat themselves with medication without going to a physician for a prescription, it is called self-medication (SM). This study aims to detect the prevalence rate of SM among ophthalmic patients in Jordan. **Methods:** This study applied a cross-sectional study design using a questionnaire tool completed by 1009 ophthalmic patients. Binary logistic regression was used to identify predictors of practicing SM. **Results:** The prevalence rate of participants practicing SM for ophthalmic conditions was 21.0%. The most common reason for SM was medical recommendations from a pharmacist or optician (55.7%), followed by failure to recognize the severity of the symptoms so that the patients thought they did not need a doctor to treat their eyes (34.4%). Eye symptoms included redness (69.3%), itching (61.3%), and a burning sensation (38.7%). The 24–30 age group was more likely to practice SM (*p* < 0.05). Other patients who had previously undergone ophthalmic surgery and were currently wearing contact lenses were less susceptible to SM practice (*p* < 0.05). **Conclusions:** Younger patients with ophthalmic conditions showed a higher probability of practicing SM. SM for eye diseases carries significant risks and requires careful consideration to avoid harmful consequences.

## 1. Introduction

One approach patients use for disease management and pain relief is self-medication (SM), which is defined as the use of medicinal products by the consumer to treat self-recognized disorders or symptoms [1], the intermittent or continuous use of prescription medications for chronic or recurring illnesses or symptoms [2,3,4], or the behavior to which patients resort to treat and cure themselves by themselves, without an official prescription [5]. Patients who use SM may not have considered the dangers of resorting to this behavior and consider it safe [6]. Self-medication is based on self-diagnosis of the disease and symptoms; repeating a previous prescription without a physician’s referral or using leftover medications also falls under SM. For ophthalmic patients, SM involves purchasing pharmaceutical products such as eye drops or other ophthalmic preparations without consulting an ophthalmologist [7] to relieve unwanted eye symptoms such as discomfort and itching [8]. Therapy is not only limited to conventional treatment; it could also include homemade products [9]. SM occurs all over the world [10].

Recently, SM has become extremely popular in many countries, with a wide variety of diseases and medications. The prevalence rates are as follows: Mexico, Italy, and Switzerland (8%), Spain, the UK, and Sweden (9%), Germany and Australia (11%), the USA (13%), and KSA (14%) [11]. SM is not exclusive to ophthalmic conditions and can involve many different drugs. A recent study in Jordan found that the prevalence of SM among the general public has reached 57.0% [12]. A study of the Jordanian population found that 39.5% of respondents wanted to self-medicate with antibiotics without paying a doctor’s consultation fee. Patients’ choices to take antibiotics were linked to several factors, including income, level of education, and age. The study participants acknowledged that their reason for SM was the availability of the drug at home from previous use, which meant that buying a new drug was unnecessary. The most popular classes of antibiotics used were penicillin (70.7%), cephalosporins (12.7%), macrolides (7.5%), quinolones (3.6%), tetracyclines (1.2%), and sulfonamides (1.1%) to treat sore throat, runny nose, and flu [13].

The prevalence of ophthalmic SM is 25.6–73.6% [14]. This is a popular practice all over the world and an indicator of how easy it is to access over-the-counter (OTC) drugs. Dealing with eye problems is risky because the eye is a sensitive organ, especially for patients who self-diagnose and self-medicate without consulting an ophthalmologist. This can lead to many eye complications due to the potential for choosing unsuitable medications, which may cause drug-drug and drug-disease interactions. The most dangerous risk is blindness. Furthermore, the risks of SM are greater for diabetic patients who suffer from eye diseases. This study aimed to explore the prevalence of SM, identify its predictors, identify ophthalmic conditions for which patients self-medicate, and describe the reasons behind this practice, especially among patients in Jordan with ophthalmic conditions. This study is unique because no previous studies about SM for ophthalmic conditions have been conducted in Jordan.

The spontaneous reporting of adverse reactions to authorities is essential for the post-marketing safety evaluation of medications [15]. Under-reporting is an issue, with only 3% of all adverse reactions that occur being reported [15]. The use of medications necessitates a balance between their inherent risks and their therapeutic advantages [16]. The perception of risk can significantly influence patient behavior, as patients who perceive a high risk of adverse drug reactions may refuse to take their medications or may hesitate to initiate new medications [16].

## 2. Materials and Methods

### 2.1. Study Design

Between 5 January and 9 March 2023, a prospective cross-sectional survey study of patients with ophthalmic conditions used face-to-face interviews and a structured questionnaire. The study was conducted at the National Center for Diabetes, Endocrinology, and Genetics (NCDEG) in Amman, Jordan. Affiliated with the Higher Council for Science and Technology, it includes almost all clinics dealing with diabetes, such as diabetic foot care clinics, pediatric diabetes, diabetes and endocrinology, a heart clinic, an eye clinic, and a genetics laboratory.

### 2.2. Data Collection

The study questionnaire tool developed by the research team was based on literature reviews [6,7,14]. The data collected included the patient’s age, gender, marital status, monthly income, level of education, and other ocular medical conditions they suffered from. In addition, patients were asked about any comorbidities, their previous ophthalmic surgery history (such as cataract surgery, refractive surgery, vitrectomy, glaucoma surgery, strabismus surgery, corneal transplants, orbital surgery, or retinal laser treatment), whether they wore contact lenses, whether they had self-medicated during the past year, which medications they used for SM, and the reasons and motives behind SM. The patients were also asked whether their symptoms improved or the problem worsened after self-medicating and what symptoms prompted them to self-medicate.

### 2.3. Ethical Approval

Ethical approval for this study was obtained from the Scientific Research Ethics Committee of Isra University, Amman, Jordan (SREC/22/12/65) and the Research Ethics Committee of the NCDEG, Amman, Jordan. All the patients consented to participate in the study before their participation.

### 2.4. Sampling Method

This study used a convenience sampling technique to recruit the study sample. All patients with ophthalmic conditions at the participating healthcare center were invited to participate in the study after being informed about its aims and objectives and giving their consent. The inclusion criteria were patients with ophthalmic conditions aged 18 years and over. Patients who consented to participate in this study were invited to participate. For patients with limited literacy or understanding of the study’s purpose, informed consent was obtained through oral explanation and their caregivers. Patients were informed that their data would be saved in a password-protected computer accessed only by the principal investigator, that no personal data would be collected for this study, and that their data would be anonymized.

### 2.5. Sample Size

The target sample size was estimated using a population size of more than 866,500 individuals diagnosed with diabetes mellitus in Jordan [17]. With a confidence interval of 95%, a standard deviation of 0.5, and a margin of error of 5%, the minimum sample size required was 379 patients.N = (Z^2^*p^(1 − p)^)/ε^2^
where Z is the Z-score, N is the population size, p is the population proportion, and ε is the margin of error.

### 2.6. Statistical Analysis

Data were analyzed using SPSS software, version 27. Categorical data were reported as percentages (frequencies). Binary logistic regression was used to identify predictors of practicing SM. The dummy variable for logistic regression was defined as confirming the practice of SM without consulting a physician. The level of significance assigned was 5%.

## 3. Results

### 3.1. Patients’ Sociodemographic Characteristics

The total number of patients was 1009. Females constituted 63.5% of the study sample. Over two-thirds of the sample (69.2%) were aged 51 and over. The majority (69.3%) of the patients were married. Only 8.5% had higher education. Nearly two-fifths (39.0%) of the patients were unemployed. The monthly family income of 62.6% of the patients was less than JOD 500. Three-quarters (73.4%) of the patients who filled out the questionnaire had comorbidities. More than half (59.9%) of the participants with accompanying ophthalmic conditions were patients with diabetes mellitus. Of the ophthalmic conditions the patients were currently suffering from, the most common eye disease was a refractive error (76.0%). Table 1 presents the sociodemographic characteristics of all the participants.

### 3.2. SM Profile of the Patients

The research included patients’ SM practices after estimating the prevalence rate of this behavior, especially for patients with diabetes in conjunction with eye diseases. The prevalence rate was 21.0%. To explore patients’ SM profiles, the questionnaire included specific questions, which first asked the patient whether they were currently using or had ever used ophthalmic medication for any condition without a prescription or without consulting a specialist (ophthalmologist) and whether the person advising them regarding SM was a pharmacist, nurse, friend, relative, or other. A total of 21.0% of the patients were defined as practicing SM.

Based on the patients’ answers concerning what categories of drugs they used for SM and what eye diseases they were suffering from before they self-medicated, two classes of medications were used: natural, such as ayurvedic or herbal (34.4%), and pharmaceutical products, including artificial tears (27.4%), Figure 1. These different classes of drugs were used to treat a variety of eye ailments that led to the adoption of SM; the most common patient complaints were eye redness (69.3%) and eye itching or a gritty sensation (61.3%) (Table 2).

### 3.3. Factors and Reasons Affecting SM

Over half (55.7%) of the patients reported that the reason for SM was advice from a pharmacist, optician or compounder; one-third (34.4%) considered their symptoms insufficiently severe to need consultation with a doctor (Table 3).

### 3.4. Side Effects in Participants After Ophthalmic SM

The most commonly reported side effects following ophthalmic SM were redness (4.7%), itching (3.8%), blurred vision (2.8%), bad sensation (1.9%), swelling (1.4%), eye secretions and discharge (1.4%), and others, Figure 2.

### 3.5. Predictors of SM Practice Among Diabetic Patients with Ophthalmic Conditions

Referring to the age groups of the participants, the binary logistic regression analysis determined that the 24–30-year-old group was more likely to practice SM (*p* < 0.05). Patients who had previously undergone ophthalmic surgery and those currently wearing contact lenses were less susceptible to SM (*p* < 0.05) (Table 4), Figure 3.

## 4. Discussion

This study examined SM practices in 1009 patients with ophthalmic conditions. More than half of the study sample (59.9%) considered that diabetes mellitus induces the development of ophthalmic diseases, especially with uncontrolled blood glucose levels. The unique point is that diabetes mellitus was not limited to a specific group. In this study, the prevalence rate of SM among patients with ophthalmic conditions was 21%. This is lower than the SM rates reported in previous studies in Nigeria 73.6% [18], Saudi Arabia 62.4% [6], Tanzania 59.8% [19], India 41.2% [10], Brazil 40.5% [9], India 29.0% [5], Colombia 25.7% [20], Argentina 25.6% [7], and Northeast Ethiopia 28.6% [14]. The prevalence rates differ not only within countries but also between cities in a country, for example, 54.1% in the Riyadh region and 35% in Taif [21,22]. These numbers show how the ease of access to medications, including OTC medications, has recently increased dramatically. The prevalence of SM among patients with ophthalmic conditions in Jordan is low compared to some countries, which may reflect differences in healthcare accessibility, awareness, or cultural attitudes. In Jordan, SM practices are influenced by cultural norms, regulatory policies, and healthcare system dynamics. Culturally, health behaviors are significantly influenced by familial advice and reliance on community-based knowledge. Although the healthcare system is relatively accessible, patients may resort to SM to avoid consultation fees or extended wait periods. In Jordan, regulatory policies restrict the availability of over-the-counter medications to some extent; however, enforcement challenges may still permit the purchase of specific medicines without prescriptions. Reducing healthcare costs while expanding healthcare coverage and insurance and awareness of the proper use of medications could decrease the prevalence of improper SM and its associated complications.

The availability of over-the-counter medications in Jordan facilitates SM for various conditions, including ophthalmic diseases, and a lack of emphasis on and control over dispensing medicines. However, patients should buy medicine only when needed to eliminate unnecessary drug use. In addition, healthcare providers must tighten control over dispensing medicines and only necessary ones should be dispensed, thus reducing demand. In our study, the binary logistic regression analysis identified that the 24–30 age group was more likely to self-medicate (*p* < 0.05). Patients with previous eye surgery and those wearing contact lenses were less susceptible to SM behavior (*p* < 0.05). This shows that this age group is important in determining medication-use practices. The younger age group exhibited higher rates of SM, suggesting a potential inverse relationship between age and SM practices, unlike patients whose medical history included eye operations. This group of patients was more careful with their treatment methods. Eye disease history had a lower influence on SM.

Concerning patients’ reasons and motivations behind self-medicating, the most commonly reported factor was getting advice from a pharmacist, optician, or compounder (55.7%), followed by the belief that the symptoms were minor (34.4%). The reasons that emerged from our study showed that heeding a pharmacist’s advice seemed to be the main reason for SM, which could lead to raising the safety margin of medication selection compared to obtaining information from invalid or low-quality sources because pharmacists have a great influence on the choice of medication for the treatment of patients [23]. This does not discourage consulting an ophthalmologist when the ailment calls for it but reduces risks in other cases. On the other hand, failing to determine whether the symptoms are severe and assuming they are not may mask the problem and lead to wrong treatment choices because the previous evaluation was ineffective and the ailment has no accurate diagnosis.

Factors associated with SM depend on the population, country, and environment. The reasons behind SM are critical because they directly affect the decision-making process. This contributes to the practice of SM and differs from one environment to another. Northeast Ethiopia demonstrated that, unlike Jordan, the main reason influencing SM was the distance of the doctor or clinic from home, followed by being too busy to see the doctor [14]. If these factors are managed well by having a healthcare center nearby, sufficient time to visit a doctor, and the belief that the symptoms are not simple, it may dissuade the patient from self-medicating. In Saudi Arabia, the motivations were different, as the highest factor was having had the same symptoms before, but ignoring the severity of the symptoms is the same as our result [6]. This disorder indicates that the Jordanian population differs from the Saudi population, showing how environments and population differences play a role in setting priorities. Furthermore, the patient’s belief that they have a sufficient medical background that qualifies them to treat themselves alone by guessing the right medicine to use was the most common reason in Argentina. The second most common reason was receiving information from family and friends [7], which again illustrates how the causes differ from one population to another. At the same time, the reasons for SM are similar, but in a different order.

In our study, the medications used by patients to self-medicate can be divided into two classes: pharmaceutical and non-pharmaceutical eye medications. One-third (34.4%) of the medicines used were herbal products, making them the most commonly used products. The second-highest percentage was artificial tears (27.4%), followed by antiallergics with or without artificial tears (25.9%). Herbal treatment is considered natural and much of the population believes in alternative medicine. This might be because herbal medicine has few side effects and is safer. Patients who self-medicate with herbal products may perceive these treatments as safer alternatives to pharmaceuticals, though this perception warrants further investigation. The least popular medication used was steroids, as only 1.4% of patients used corticosteroids. As the results show, patients with ophthalmic conditions avoided steroid eye drops and preferred to use natural medication. The low percentage of patients using steroids in this study reduces the possibility of drug-disease interaction. For example, in the case of glaucoma, patients who self-medicate with steroids will suffer grave consequences if they are not aware of contraindications [24]. Previous research in Saudi Arabia highlighted that the majority of patients (86.6%) had limited awareness concerning possible ocular complications due to the use of steroids [21,25].

The high use of antibiotics in our study may be justified because patients believe bacterial eye infection is the main cause, reminding us of a major problem, the generation of resistant bacteria. Compared to the Nigerian study [18], Jordanian patients have less chance of generating resistant bacteria because the rate of antibiotic use to self-medicate eye diseases is lower. Products used for SM were not similar across countries. For example, in Saudi Arabia, patients used the holy water Zamzam and human milk from lactating mothers [6,26], whereas in rural India they used food including lemon and oils [27].

Based on the definition used in this study, patients applied SM to treat, cure, and relieve eye symptoms. The ophthalmic symptoms complained about by all patients in this study who self-medicated (21.0%) were itching and other unpleasant sensations. All patients except one used eye drops for SM for the presence of real eye disease or symptoms with existing signs. One person self-medicated with no noticeable symptoms. Using unnecessary medication is unjustified when no sign of eye disease is apparent. However, using SM in the absence of symptoms might be for prevention or protection from future symptoms. For example, dry eye prevention could be an extra benefit of SM, but it requires a wide knowledge and background to distinguish the correct time and choice of eye drops.

Our study aimed to investigate the risks and side effects of SM problems. In this study, for one group of patients who self-medicated to relieve their painful eye symptoms, SM exacerbated their symptoms, as 4.7% had redness as a side effect. After using eye medication, 2.8% of the patients reported that their vision blurred, 3.8% experienced itching, and others had swelling or discharge. In addition, 10.8% of the patients suffered from other symptoms that might be severe. Unfortunately, in SM, any mistake, even a small one, might have serious consequences leading to unwanted irreversible conditions such as blindness [6]. Side effects could be due to contraindications, incorrect self-diagnosis, and selection of inappropriate medication [4].

To reduce risks and maximize the benefits of SM, the following measures should be implemented [28]: strengthening surveillance systems, encouraging spontaneous notification of SM to pharmacovigilance bodies, establishing patient-physician-pharmacist partnerships to educate and inform the general public about SM, and increasing awareness of the risks and dangers of misuse and abuse. Health professionals should be the primary source of guidance for the general public regarding SM behaviors [28].

Despite the risk associated with inappropriate SM, when used appropriately, SM can provide numerous advantages to both individuals and health systems. These benefits include reductions in time spent waiting in line for medical appointments, physicians’ workloads, healthcare costs, and absenteeism from work, and the prevention of the allocation of limited medical resources to minor conditions [29]. Safer SM practices could be encouraged through proper training and education and by encouraging pharmacists to inform patients about drug use during consultations while enforcing stricter regulations on dispensing practices.

### Study Limitations

The design of our study is cross-sectional, so no follow-up of patients who self-medicated their ophthalmic conditions occurred. Moreover, the cross-sectional design does not allow for causal inferences or long-term insights into the effects of SM practices. Also, a self-administered questionnaire study design is prone to social desirability bias, in which the patients might answer the questions inaccurately to reflect good attitudes and practices. Using anonymous surveys or validated instruments can minimize this type of bias. Furthermore, this is a single-center study, which might restrict the generalizability of our study findings. The use of a convenience sampling technique might have affected the generalizability of our study findings, as the demographic characteristics of the study participants (such as overrepresentation of older individuals (51 years and above)) were not representative of the whole targeted study population. Moreover, no previous studies have examined SM among diabetic patients with ophthalmic conditions, which limited the comparison of the findings to similar study populations. Therefore, our study findings should be interpreted carefully.

## 5. Conclusions

SM has become a widespread global phenomenon and the Arab world is no exception. It is not limited to a particular age group or specific diseases. Even eye diseases have become common and most medications are OTC, which shows the ease of access to medicines in the Jordanian market due to their abundance and the absence of strict control on dispensing medications without a prescription by health care providers. This continuously increases the demand for medicines due to their rapid availability. Self-medication has both risks and if applied correctly, benefits, but even when carried out correctly, caution must be taken to avoid any consequences, especially when dealing with eye diseases. Future multi-center studies are warranted to enhance the generalizability of the study findings. These studies should have a more balanced age distribution to explore age-related trends and adopt a longitudinal study design to examine behavioral trends in SM practices over time.

## Figures and Tables

**Figure 1 healthcare-13-00372-f001:**
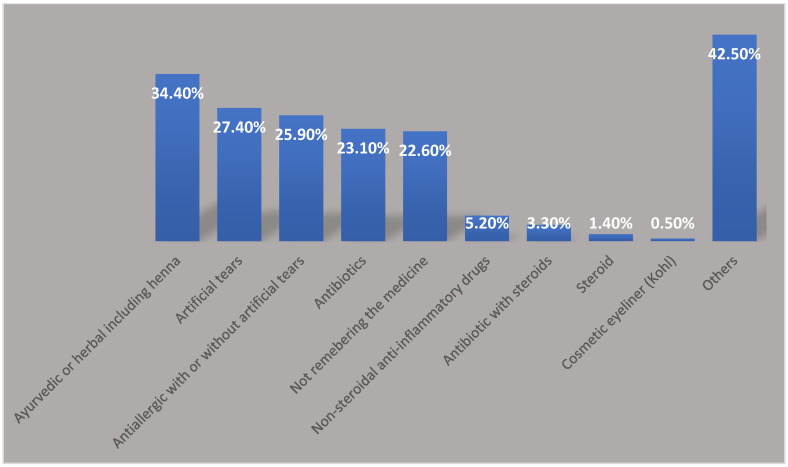
Medication categories used in SM.

**Figure 2 healthcare-13-00372-f002:**
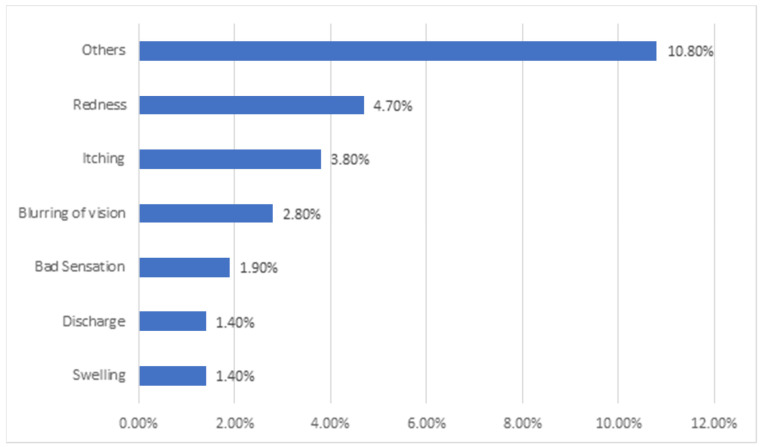
Side effects in participants after ophthalmic SM practice.

**Figure 3 healthcare-13-00372-f003:**
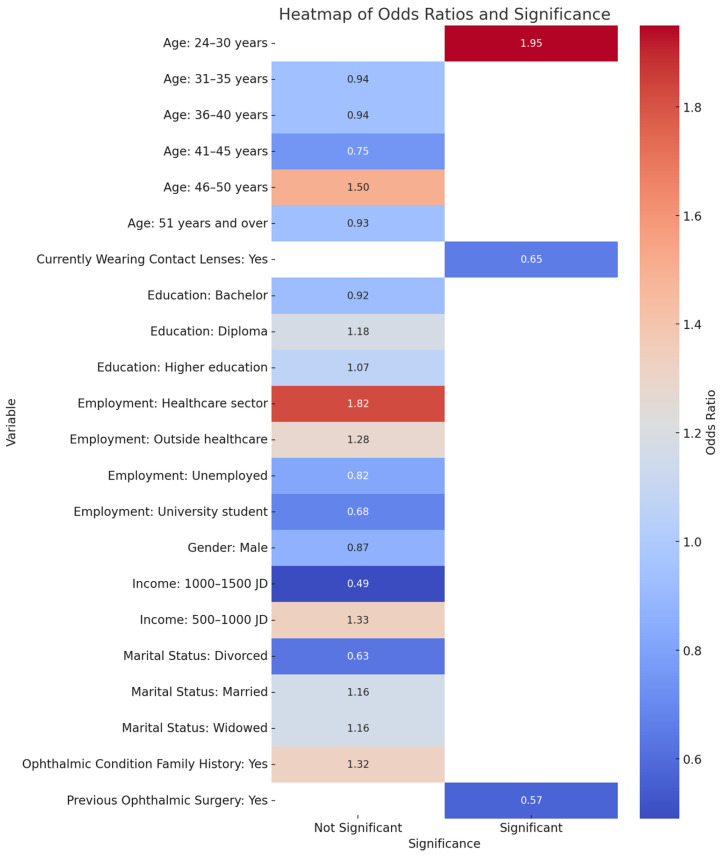
Heatmap for predictors of SM.

**Table 1 healthcare-13-00372-t001:** Baseline sociodemographic characteristics of the patients.

Variable	Frequency	Percentage
**Gender**		
Female	641	63.5%
**Age Group ***		
18–25 years	83	8.2%
26–30 years	48	4.8%
31–35 years	25	2.5%
36–40 years	40	4.0%
41–45 years	36	3.6%
46–50 years	79	7.8%
51 years and over	698	69.2%
**Marital Status**		
Single	187	18.5%
Married	699	69.3%
Divorced	7	0.7%
Widowed	116	11.5%
**Education**		
High school or lower	389	38.6%
Bachelor degree	367	36.4%
Higher education (master’s degree or PhD)	86	8.5%
Diploma	167	16.6%
**Employment Status**		
Retired	325	32.2%
Unemployed	398	39.4%
Work in the healthcare sector	28	2.8%
University student	45	4.5%
Work outside the healthcare sector	213	21.1%
**Monthly Income of the Family**		
Less than JOD 500	632	62.6%
JOD 500–1000	338	33.5%
JOD 1000–1500	34	3.4%
JOD 1500 and above	5	0.5%
**Presence of Comorbidities**		
Yes	741	73.4%
**Comorbidities (Any Chronic Diseases Co-Existing with the Ophthalmic Conditions)**		
**Endocrine, Nutritional, and Metabolic Diseases**		
Diabetes mellitus	604	59.9%
Dyslipidemia	306	30.3%
**Diseases of the Circulatory System**		
Hypertension	461	45.7%
Cardiovascular diseases	77	7.6%
**Diseases of the Musculoskeletal System and Connective Tissue**		
Osteoarthritis	43	4.3%
Rheumatoid arthritis	40	4.0%
Others	147	14.6%
**Ophthalmic Conditions (Any Eye Diseases the Patient Is Currently Suffering From)**		
Allergic conjunctivitis	28	2.8%
Refractive error	767	76.0%
Cataract	289	28.6%
Glaucoma	81	8.0%
Conjunctival masses	6	0.6%
Vitreoretinal disorders	103	10.2%
Viral conjunctivitis	22	2.2%
Bacterial conjunctivitis	4	0.4%
**Previous Ophthalmic Surgery**		
Yes	300	29.7%
**Ophthalmic Condition Family History (Does Any Family Member Have an Ophthalmic Disease such as Glaucoma or Use Eyeglasses?)**		
Yes	804	79.7%
**Currently Wearing Contact Lenses**		
Yes	191	18.9%

* The first age group is for people between 18 and 25 years old and covers 7 years. The next groups each cover 4 years. This was carried out on purpose to make sure there were enough young people in the study, since fewer people aged 18–21 took part. To keep our research and comparisons consistent, we chose to use a 4-year gap for the next groups. The decision to use a 4-year interval instead of 5 years was made because shorter intervals, like 4 years, help to keep groups more consistent and lower differences within each category.

**Table 2 healthcare-13-00372-t002:** Eye conditions that led patients to ophthalmic SM (n = 212).

Variable	Frequency	Percentage
Redness of eyes	147	69.3%
Eye itching/gritty sensation	130	61.3%
Photophobia/glare	94	44.3%
Burning sensation	82	38.7%
Eye pain/strain	81	38.2%
Watery eyes	72	34.0%
Discharge from eyes	49	23.1%
Defective vision	19	9.0%
No symptoms	1	0.5%
Others	77	36.3%

**Table 3 healthcare-13-00372-t003:** Factors and reasons affecting SM.

Variable	Frequency	Percentage
Advice from a pharmacist, optician, or compounder	118	55.7%
Thought symptoms were not severe	73	34.4%
Advice from family and friends	61	28.8%
Had had the same symptoms before	42	19.8%
Lacked the time to see a doctor	33	15.6%
Disliked long waiting times at hospitals or clinics	31	14.6%
High cost of treatment or not covered by health care insurance companies	26	12.3%
Long distance to eye care services (the hospital or clinic was far from their home)	20	9.4%
Thought themselves capable of self-treatment	13	6.1%
Had some medication at home (opened)	9	4.2%
Bad image of medical service due to previous experience	7	3.3%
Had some medication at home (sealed)	7	3.3%
Others/no reason	72	34.0%

**Table 4 healthcare-13-00372-t004:** Binary logistic regression analysis for patients’ sociodemographic characteristics.

Variable	Odds Ratio of SM (95% Confidence Interval)	*p*-Value
**Gender**		
Female (reference category)	1.00	
Male	0.87 (0.63–1.20)	0.393
**Age Groups**		
18–23 years (reference category)	1.00	
24–30 years	1.95 (1.05–3.63)	0.035 *
31–35 years	0.94 (0.35–2.53)	0.900
36–40 years	0.94 (0.43–2.07)	0.873
41–45 years	0.75 (0.31–1.81)	0.516
46–50 years	1.50 (0.90–2.52)	0.122
51 years and over	0.93 (0.67–1.29)	0.657
**Marital Status**		
Single (reference category)	1.00	
Married	1.16 (0.83–1.62)	0.390
Divorced	0.63 (0.08–5.22)	0.664
Widowed	1.16 (0.73–1.84)	0.525
**Education**		
High school or lower (reference category)	1.00	
Bachelor’s degree	0.922 (0.67–1.27)	0.617
Higher education	1.07 (0.63–1.83)	0.797
Diploma	1.18 (0.79–1.75)	0.416
**Employment Status**		
Retired (reference category)	1.00	
Unemployed	0.82 (0.60–1.13)	0.228
Work in the healthcare sector	1.82 (0.81–4.07)	0.148
University student	0.68 (0.30–1.55)	0.361
Work outside the healthcare sector	1.28 (0.90–1.84)	0.171
**Monthly Income for the Family**		
Less than JOD 500 (reference category)	1.00	
JOD 500–1000	1.33 (0.97–1.82)	0.073
JOD 1000–1500	0.49 (0.17–1.41)	0.187
JOD 1500 and above	**-**	**-**
**Previous Ophthalmic Surgery**		
No (reference category)	1.00	
Yes	0.57 (0.40–0.82)	0.002 **
**Ophthalmic Condition Family History (Does Any Family Member Have An Ophthalmic Disease such as Glaucoma or Use Eyeglasses?)**		
No (reference category)	1.00	
Yes	1.32 (0.89–1.96)	0.175
**Currently Wearing Contact Lenses**		
No (reference category)	1.00	
Yes	0.65 (0.43–0.99)	0.047 *

*p*-Value: * *p* < 0.05; ** *p* < 0.01.

## Data Availability

The datasets used and/or analyzed during the current study are available from the corresponding author on reasonable request.

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
