# Peer review of "Prevalence and Predictors of Self-Medication Among Ophthalmic Patients in Jordan: A Cross-Sectional Analysis"

_healthcare, 2025, doi:10.3390/healthcare13040372_

Round 1

Reviewer 1 Report

Comments and Suggestions for Authors

pl. check the file enclosed.

Comments on the Quality of English Language

The opinion of other reviewers should be considered in this regard. 

Author Response

Reviewer 1:

The paper addresses problem of self-medication for ophthalmic conditions in a

developing economy, which is relevant from the point of view of health care.

  1. In my opinion, the message of the paper will be conceived by the reader with

difficulty. The English language is to be fine-tuned (my native language is not English,

hence opinion of other reviewers in this regard should be considered accordingly).

- Thank you for this comment. We have now addressed this point and checked the language of the study.

  1. It could be of value if authors include the questionnaire in the text as a table or as an

appendix.

- Thank you for this comment. We have now addressed this point and added the questionnaire tool as appendix.

  1. Line – 25-26, 88. What type of surgery the authors expected should be clarified by

the respondents?

- Thank you for this comment. We have now clarified this point in the manuscript as ophthalmic surgery history included cataract surgery, refractive surgery, vitrectomy, glaucoma surgery, strabismus surgery, corneal transplants, orbital surgery, or retinal laser treatment, see lines 104-107.

  1. Once self-medication has been abbreviated as SM in the text, it should be used

everywhere in the text following.

- Thank you for this comment. We have now addressed this comment in the manuscript throughput the manuscript.

  1. Line 62 – OTC has not been defined previously.

- Thank you for this comment. We have now addressed this comment in the manuscript, see line 67.

  1. The aim of the study was to explore the prevalence and identify the predictors of

practicing SM, what was done after completing the study? This point is not critical,

but it may add scientific weightage to the study done.

- Thank you for this comment. This study has identified high risk population for SM practices and provided recommendations for future research. We have now highlighted these points in the manuscript and in the conclusion.

  1. Another point I would like to have explanation is that the study was done at a

specific center, dealing with diabetic patients. That means that population sample

which took part in the study were from a specific group. But the sample size was

calculated from the total population of Jordon. Pl. specify.

- Thank you for this comment. We have now addressed this comment and recalculated the sample size for patients with diabetes in Jordan. The National Centre for Diabetes, Endocrinology, and Genetics (NCDEG) in Amman, Jordan is the largest centre for diabetes management in Jordan. Patients from all over the country refer to this centre. The number of annual visits for this centre exceed 500,000 visits every year. Based on the International diabetes Federation, the estimated number of Jordanians with diabetes mellitus is around 866,500 individuals, therefore, with a confidence interval of 95%, a standard deviation of 0.5, and a margin of error of 5%, the minimum required sample size was 379 patients, see lines 131-137.

  1. Line 85- pl. start the sentence with capital letter.

- Thank you for this comment. We have now addressed this comment, see line 101.

10.Paragraph (line 120-127), pl. check English, some meanings are not clear.

- Thank you for this comment. We have now addressed this comment.

11.Table 1 –

  1. the age groups defined in the table differ from each other: 18-23 years – 5

years; 24-30 years – 6 years; 31-35 years – 4 years and so on. Pl. make

changes accordingly.

- Thank you for this comment. We have now addressed this comment, see Table 1.

  1. What difference do authors meant when they described “bachelor’s degree

and higher education” etc. pl. clarify.

- Thank you for this comment. We have now clarified that higher education means (Master degree and PhD), see Table 1.

  1. c. The authors should retabulate the data of table 1. Specify the comorbidities

as per the international classification of diseases. That will be more

presentable.

- Thank you for this comment. We have now addressed this comment, see Table 1.

12.Figure 1. The percentage values have been given twice in the same graph. Pl. check.

- Thank you for this comment. We have now addressed this comment and replaced the figure, see Figure 1.

13.Line 156 -meaning not clear.

- Thank you for this comment. We have now addressed this comment and clarified the sentence.

14.Lines – 171-178. The authors compare their results with results of similar type of

studies from other countries. The comparison in my opinion is not suitable, the

authors studied specific group of patients from diabetic clinic, where the mentioned

studies dealt with other types of groups e.g. reference 5-7 and others.

- Thank you for this comment. In light of the absence of any prior research that investigated self-medication among diabetic patients with ophthalmic conditions, we preferred to compare our results with those of comparable studies that have addressed self-medication in other patient populations. This approach was implemented to emphasize the novelty of our research by concentrating on this particular subgroup and to provide context. Besides, based on the reviewer comment, we have now added this point to the study limitations, see lines 373-376.

15.Very detailed discussion – this can be reduced in size to make it more appropriate.

- Thank you for this comment. We have now addressed this comment as appropriate.

16.Line 242 – pl. check the English.

- Thank you for this comment. We have now addressed this comment, see page 10.

17.In conclusion – pl. specify the findings/results as per the aim/aims of the study.

- Thank you for this comment. We have now addressed this comment and revised the conclusion.

Overall, the manuscript needs major revision.

Thank you for your valuable comments. We have now addressed all comments as highlighted in the manuscript.

Reviewer 2 Report

Comments and Suggestions for Authors

The manuscript, "Self-Medication Among Patients with Ophthalmic Conditions in Jordan: A Cross-Sectional Study," performs a study on a very important topic for the safety of medicines.

Title - Options for Improvement

The title is an important issue in an article and is the “face” of the research. For that reason, look for the two options and reasoning,

1.    Highlighting the Core Findings

o  "Prevalence and Predictors of Self-Medication Among Ophthalmic Patients in Jordan: A Cross-Sectional Analysis"

2.    Emphasizing the Risks and Context

o  "Self-Medication Practices and Associated Risks Among Ophthalmic Patients in Jordan: A Cross-Sectional Study"

Limitations

1.    Introduction

o  Regarding the type of sample there is a lack of information about patient's engagement with pharmacovigilance and adverse drug reactions.

2.    Study Design and Generalizability

o  The study is limited to a single health center, which restricts the generalizability of the findings to the broader population in Jordan or similar settings.

o  The cross-sectional design does not allow for causal inferences or long-term insights into the effects of self-medication practices.

3.    Data Collection Bias

o  The reliance on self-reported data through questionnaires may introduce social desirability bias, where participants could provide responses perceived as socially acceptable rather than truthful.

4.    Sample Representation

o  While the sample size exceeds the minimum requirement, the overrepresentation of older individuals (51 years and above) may skew the findings, as this age group may differ significantly in self-medication behaviours compared to younger cohorts.

5.    Lack of Comparative Context

o  The manuscript mentions prevalence rates from other countries but does not delve deeply into why these differences exist or how they inform the findings in Jordan.

6.    Discussion of Risks

o  Although risks of self-medication are highlighted, there is limited emphasis on specific interventions or strategies to mitigate these risks, such as public health campaigns or regulatory changes.

7.    Visual Data Representation

o  Figures and tables are underutilized for summarizing key findings, such as the predictors of self-medication or side effects.

8.    Ethical Considerations

o  While ethical approval is mentioned, there is no detailed discussion of how participant confidentiality and data security were ensured. Also, state if the consent was informed.

Recommendations for Improvement

  1. Introduction

o  Consider the introduction of the topic of patients' engagement with ADR and risk perception increasing also the broad of references. If you agree look for the following suggestions.

·       Joaquim, J.J., Matos, C., Mateos-Campos, R. Assessment of risk perception of patients concerning adverse drug reactions. Curr. Issues Pharm. Med. Sci., 2023, 36(2), pp. 103–107. https://doi.org/10.2478/cipms-2023-0018

·       Nichols, V., Thériault-Dubé, I., Touzin, J. et al. Risk Perception and Reasons for Noncompliance in Pharmacovigilance. Drug-Safety 32, 579–590 (2009). https://doi.org/10.2165/00002018-200932070-00004

  1. Expand Generalizability

o  Consider including multiple centers or regions in future studies to enhance the representativeness of the findings.

  1. Mitigate Bias

o  Employ strategies to minimize social desirability bias, such as anonymous surveys or validated instruments.

  1. Broaden Age Representation

o  Ensure a more balanced age distribution to explore age-related trends in self-medication practices comprehensively.

  1. Enhance Comparative Analysis

o  Provide deeper analysis comparing Jordanian practices with those in other countries, focusing on cultural, regulatory, and healthcare system differences.

  1. Propose Interventions

o  Suggest concrete measures, such as pharmacist training or stricter regulation of OTC drugs, to address the identified issues.

  1. Improve Data Visualization

o  Include additional charts or graphs to make complex data more accessible, such as a heatmap for predictors of self-medication or a flowchart of common self-medication pathways.

  1. Ethical Transparency

o  Expand on the ethical safeguards implemented during the study, such as data anonymisation and informed consent processes.

Some questions to clarify some topics

Study Design and Methodology

1.    Generalizability

o  How do you plan to address the limited generalizability of the findings due to the single-center study design? Would you consider including multiple centers in future studies?

2.    Bias Mitigation

o  What measures were taken to reduce social desirability bias in participants’ responses to the self-administered questionnaire?

3.    Sampling

o  Given the overrepresentation of older participants, how might this affect the applicability of your findings to younger populations? Could you explore age-stratified analyses?

Data Presentation

4.    Visual Representation

o  Could you include additional figures or tables to summarize key findings, such as predictors of self-medication or the prevalence of specific symptoms leading to self-medication?

5.    Comparative Context

o  Can you provide a more detailed comparison of your findings with similar studies in other regions or countries? What cultural or systemic factors might explain the differences?

Discussion and Interpretation

6.    Intervention Strategies

o  Could you propose specific public health interventions or policy recommendations to address the risks associated with self-medication in Jordan?

7.    Risks and Benefits

o  While the risks of self-medication are discussed, could you elaborate on the potential benefits and how they can be maximized through proper education or regulation?

Ethical Considerations

8.    Participant Confidentiality

o  How was participant confidentiality ensured during data collection and analysis? Were any additional safeguards implemented?

9.    Consent Process

o  Could you elaborate on how informed consent was obtained, particularly for participants with limited literacy or understanding of the study’s purpose?

Limitations and Future Directions

10.     Study Limitations

o  The study mentions the cross-sectional design as a limitation. Could you suggest alternative study designs that might provide more robust insights, such as longitudinal studies?

11.     Future Research

o  What additional research questions have emerged from your findings, and how do you plan to address them in subsequent studies?

Context and Relevance

  1. Cultural Factors

o  Could you discuss in greater detail how cultural attitudes towards healthcare and self-treatment in Jordan might influence self-medication practices?

  1. Healthcare System

o  How do you think gaps in the healthcare system, such as accessibility or cost, contribute to self-medication trends? Could these be mitigated through systemic changes?

Overstated Claims and Suggestions

1.    Claim

o  "Self-medication, especially in the field of eye diseases, is considered a double-edged sword that may be harmful but could also have benefits, provided that it is practiced correctly."
Issue This statement implies that self-medication can be beneficial if done correctly, but the study does not sufficiently explore or provide evidence for the "benefits" of self-medication.
Suggested Rephrase

o  "Self-medication for eye diseases carries significant risks and requires careful consideration to avoid harmful consequences."

2.    Claim

o  "The prevalence of self-medication among patients with ophthalmic conditions in Jordan is lower compared to other countries, indicating better healthcare practices or stricter regulations."
Issue The study does not provide direct evidence linking the lower prevalence to healthcare practices or regulations.
Suggested Rephrase

o  "The prevalence of self-medication among patients with ophthalmic conditions in Jordan is lower compared to some countries, which may reflect differences in healthcare accessibility, awareness, or cultural attitudes."

3.    Claim

o  "Patients who self-medicate with herbal products believe these treatments are risk-free and effective."
Issue The manuscript does not provide data on patients’ perceptions of herbal treatments being "risk-free."
Suggested Rephrase

o  "Patients who self-medicate with herbal products may perceive these treatments as safer alternatives to pharmaceuticals, though this perception warrants further investigation."

4.    Claim

o  "The younger age group was more dependent on self-medication, governing the inverse relationship between age and self-medication."
Issue The term "governing" is too strong without clear evidence of causality.
Suggested Rephrase:

o  "The younger age group exhibited higher rates of self-medication, suggesting a potential inverse relationship between age and self-medication practices."

5.    Claim

o  "The abundance of over-the-counter medications in Jordan makes it easy for patients to self-medicate for any disease at any time."
Issue The phrase "any disease at any time" is overly broad and not substantiated by the study.
Suggested Rephrase

o  "The availability of over-the-counter medications in Jordan facilitates self-medication for various conditions, including ophthalmic diseases."

General Recommendations

·     Use cautious language like "may," "suggests," or "could indicate" when discussing findings that are not directly supported by data.

·     Avoid making causal claims unless the study design and analysis provide robust evidence.

·     Ensure all statements are directly supported by the data or references cited in the manuscript.

Comments on the Quality of English Language
Recommendations for mprovement of the quality of the English language
  1. Proofreading Conduct a thorough review for grammatical errors, awkward phrasing, and redundancy.
  2. Professional Editing Consider using a professional editing service to ensure the manuscript adheres to the highest standards of academic English.
  3. Sentence Variety Vary sentence structure to avoid monotony and enhance engagement.
  4. Focus on Conciseness Remove unnecessary words or phrases to make the manuscript more succinct.

Author Response

Reviewer 2:

The manuscript, "Self-Medication Among Patients with Ophthalmic Conditions in Jordan: A Cross-Sectional Study," performs a study on a very important topic for the safety of medicines.

Title - Options for Improvement

The title is an important issue in an article and is the “face” of the research. For that reason, look for the two options and reasoning,

  1. Highlighting the Core Findings

o  "Prevalence and Predictors of Self-Medication Among Ophthalmic Patients in Jordan: A Cross-Sectional Analysis"

  1. Emphasizing the Risks and Context

o  "Self-Medication Practices and Associated Risks Among Ophthalmic Patients in Jordan: A Cross-Sectional Study"

 - Thank you for this comment. We have now addressed this comment and changed the study title to be “Prevalence and Predictors of Self-Medication Among Ophthalmic Patients in Jordan: A Cross-Sectional Analysis”.

Limitations

  1. Introduction

o  Regarding the type of sample there is a lack of information about patient's engagement with pharmacovigilance and adverse drug reactions.

- Thank you for this comment. We have now addressed this comment and highlighted this point in the introduction using the recommended references below, see lines 80-86.

  1. Study Design and Generalizability

o  The study is limited to a single health center, which restricts the generalizability of the findings to the broader population in Jordan or similar settings.

  • The cross-sectional design does not allow for causal inferences or long-term insights into the effects of self-medication practices.

 - Thank you for this comment. We have now highlighted these limitations in the manuscript, see lines 362-376.

  1. Data Collection Bias

o  The reliance on self-reported data through questionnaires may introduce social desirability bias, where participants could provide responses perceived as socially acceptable rather than truthful.

- Thank you for this comment. We totally agree with the reviewer concerning this common limitation among self-reported data, this limitation is highlighted in the limitations section, see lines 365-367.

  1. Sample Representation

o  While the sample size exceeds the minimum requirement, the overrepresentation of older individuals (51 years and above) may skew the findings, as this age group may differ significantly in self-medication behaviours compared to younger cohorts.

- Thank you for this comment. We have now highlighted this point further in the study limitations section, as convenience sampling technique might affect the generalizability of our study findings, see lines 370-373.

  1. Lack of Comparative Context

o  The manuscript mentions prevalence rates from other countries but does not delve deeply into why these differences exist or how they inform the findings in Jordan.

- Thank you for this comment. In light of the absence of any prior research that investigated self-medication among diabetic patients with ophthalmic conditions, we preferred to compare our results with those of comparable studies that have addressed self-medication in other patient populations. This approach was implemented to emphasize the novelty of our research by concentrating on this particular subgroup and to provide context. Besides, based on the reviewer comment, we have now added this point to the study limitations, see lines 373-376.

  1. Discussion of Risks

o  Although risks of self-medication are highlighted, there is limited emphasis on specific interventions or strategies to mitigate these risks, such as public health campaigns or regulatory changes.

- Thank you for this comment. We have now provided recommendations to mitigate the risk of self-medication, see lines 343-349.

  1. Visual Data Representation

o  Figures and tables are underutilized for summarizing key findings, such as the predictors of self-medication or side effects.

- Thank you for this comment. We presented the side effects and medications used graphically in Figure 1 and Figure 2. The predictors of self-medication are presented in Table 4.

  1. Ethical Considerations

o  While ethical approval is mentioned, there is no detailed discussion of how participant confidentiality and data security were ensured. Also, state if the consent was informed.

 - Thank you for this comment. Informed consent details are mentioned under the subheading “Ethical approval”, see lines 115-116. We have now added further details to clarify other points to the manuscript, see lines 123-129.

Recommendations for Improvement

  1. Introduction

o  Consider the introduction of the topic of patients' engagement with ADR and risk perception increasing also the broad of references. If you agree look for the following suggestions.

  • Joaquim, J.J., Matos, C., Mateos-Campos, R.Assessment of risk perception of patients concerning adverse drug reactions. Curr. Issues Pharm. Med. Sci., 2023, 36(2), pp. 103–107. https://doi.org/10.2478/cipms-2023-0018
  • Nichols, V., Thériault-Dubé, I., Touzin, J. et al. Risk Perception and Reasons for Noncompliance in Pharmacovigilance. Drug-Safety32, 579–590 (2009). https://doi.org/10.2165/00002018-200932070-00004

- Thank you for this comment. We have now addressed this comment and highlighted this point in the introduction using the recommended references, see lines 80-86.

  1. Expand Generalizability
  • Consider including multiple centers or regions in future studies to enhance the representativeness of the findings.

- Thank you for this comment. We have now highlighted this point and provided recommendations for future research in the study conclusion, see lines 387-390.

  1. Mitigate Bias
  • Employ strategies to minimize social desirability bias, such as anonymous surveys or validated instruments.

- Thank you for this comment. We have now highlighted this limitation further in the limitations section and provided suggestions for strategies to minimize this bias, see lines 367-368.

  1. Broaden Age Representation
  • Ensure a more balanced age distribution to explore age-related trends in self-medication practices comprehensively.

- Thank you for this comment. We have now highlighted this limitation further in the limitations section, see lines 370-373. Besides, we highlighted this point to recommendations for future research, see lines 388-389.

  1. Enhance Comparative Analysis

o  Provide deeper analysis comparing Jordanian practices with those in other countries, focusing on cultural, regulatory, and healthcare system differences.

- Thank you for this comment. We have now highlighted this point in the discussion, see lines 373-376.

  1. Propose Interventions

o  Suggest concrete measures, such as pharmacist training or stricter regulation of OTC drugs, to address the identified issues.

- Thank you for this comment. We have now provided recommendations to mitigate the risk of self-medication, see lines 343-349.

  1. Improve Data Visualization
  • Include additional charts or graphs to make complex data more accessible, such as a heatmap for predictors of self-medication or a flowchart of common self-medication pathways.

- Thank you for this comment. We have now addressed this point and added Figure 3 presenting heatmap for predictors of self-medication.

  1. Ethical Transparency
  • Expand on the ethical safeguards implemented during the study, such as data anonymisation and informed consent processes.

 - Thank you for this comment. Informed consent details are mentioned under the subheading “Ethical approval”, see lines 115-116. We have now added further details to clarify other points to the manuscript, see lines 123-129.

Some questions to clarify some topics

Study Design and Methodology

  1. Generalizability

o  How do you plan to address the limited generalizability of the findings due to the single-center study design? Would you consider including multiple centers in future studies?

- Thank you for this comment. We have now highlighted this point and provided recommendations for future research in the study conclusion, see lines 387-390.

  1. Bias Mitigation

o  What measures were taken to reduce social desirability bias in participants’ responses to the self-administered questionnaire?

- Thank you for this comment. We used anonymous survey to reduce the possibility of social desirability bias.

  1. Sampling

o  Given the overrepresentation of older participants, how might this affect the applicability of your findings to younger populations? Could you explore age-stratified analyses?

 - Thank you for this comment. The binary logistic regression analysis findings in Table 4 provided insights on the likelihood of practicing self-medication across all age groups; which showed that the age group 24-30 years were 2-folds more likely to engage in the practice compared to others.

Data Presentation

  1. Visual Representation

o  Could you include additional figures or tables to summarize key findings, such as predictors of self-medication or the prevalence of specific symptoms leading to self-medication?

- Thank you for this comment. We have now addressed this point and added Figure 3 presenting heatmap for predictors of self-medication.

  1. Comparative Context

o  Can you provide a more detailed comparison of your findings with similar studies in other regions or countries? What cultural or systemic factors might explain the differences?

 - Thank you for this comment. We have now highlighted this point in the discussion, see lines 220-232.

Discussion and Interpretation

  1. Intervention Strategies

o  Could you propose specific public health interventions or policy recommendations to address the risks associated with self-medication in Jordan?

- Thank you for this comment. We have now provided recommendations to mitigate the risk of self-medication, see lines 343-349.

  1. Risks and Benefits

o  While the risks of self-medication are discussed, could you elaborate on the potential benefits and how they can be maximized through proper education or regulation?

- Thank you for this comment. We have now addressed this point, see lines 350-359. 

Ethical Considerations

  1. Participant Confidentiality

o  How was participant confidentiality ensured during data collection and analysis? Were any additional safeguards implemented?

 - Thank you for this comment. Informed consent details are mentioned under the subheading “Ethical approval”, see lines 115-116. We have now added further details to clarify other points to the manuscript, see lines 123-129.

  1. Consent Process

o  Could you elaborate on how informed consent was obtained, particularly for participants with limited literacy or understanding of the study’s purpose?

 - Thank you for this comment. For patients with limited literacy or understanding of the study’s purpose, informed consent was obtained though oral explanation and through their caregivers. We have now highlighted this point further in the manuscript, see lines 123-129. 

Limitations and Future Directions

  1. Study Limitations

o  The study mentions the cross-sectional design as a limitation. Could you suggest alternative study designs that might provide more robust insights, such as longitudinal studies?

- Thank you for this comment. We have now provided suggestions for future research, see lines 389-390.

  1. Future Research

o  What additional research questions have emerged from your findings, and how do you plan to address them in subsequent studies?

- Thank you for this comment. Based on our study findings, we are aiming to address study limitations emerged in the current research through the implementation of different sampling technique that secure demographic balance across study participants including age distribution, besides, we are aiming to examining the behavioral trend in SM practices over time through the implementation of longitudinal study design.

Context and Relevance

  1. Cultural Factors
  • Could you discuss in greater detail how cultural attitudes towards healthcare and self-treatment in Jordan might influence self-medication practices?
  • - Thank you for this comment. We have now highlighted this point in the discussion, see lines 220-232.

  1. Healthcare System

o  How do you think gaps in the healthcare system, such as accessibility or cost, contribute to self-medication trends? Could these be mitigated through systemic changes?

- Thank you for this comment. We totally agree that limited accessibility and cost are one of the main factors that contributed to SM practices. Other contributing factors include culture and low health literacy. Reducing healthcare costs and expanding healthcare coverage and insurance and increasing awareness on the proper use of medications could decrease the prevalence of improper SM and its associated complications, see lines 220-232. 

Overstated Claims and Suggestions

  1. Claim

o  "Self-medication, especially in the field of eye diseases, is considered a double-edged sword that may be harmful but could also have benefits, provided that it is practiced correctly."
Issue This statement implies that self-medication can be beneficial if done correctly, but the study does not sufficiently explore or provide evidence for the "benefits" of self-medication.

- Thank you for this comment. We have now addressed this point, see lines 350-359. 
Suggested Rephrase

o  "Self-medication for eye diseases carries significant risks and requires careful consideration to avoid harmful consequences."

- Thank you for this comment. We have now addressed this point, see lines 30-32. 

  1. Claim

o  "The prevalence of self-medication among patients with ophthalmic conditions in Jordan is lower compared to other countries, indicating better healthcare practices or stricter regulations."
Issue The study does not provide direct evidence linking the lower prevalence to healthcare practices or regulations.
Suggested Rephrase

o  "The prevalence of self-medication among patients with ophthalmic conditions in Jordan is lower compared to some countries, which may reflect differences in healthcare accessibility, awareness, or cultural attitudes."

 - Thank you for this comment. We have now addressed this point, see lines 221-223.

  1. Claim

o  "Patients who self-medicate with herbal products believe these treatments are risk-free and effective."
Issue The manuscript does not provide data on patients’ perceptions of herbal treatments being "risk-free."
Suggested Rephrase

  • "Patients who self-medicate with herbal products may perceive these treatments as safer alternatives to pharmaceuticals, though this perception warrants further investigation."

- Thank you for this comment. We have now addressed this point, see lines 294-296.

  1. Claim

o  "The younger age group was more dependent on self-medication, governing the inverse relationship between age and self-medication."
Issue The term "governing" is too strong without clear evidence of causality.
Suggested Rephrase:

  • "The younger age group exhibited higher rates of self-medication, suggesting a potential inverse relationship between age and self-medication practices."

- Thank you for this comment. We have now addressed this point, see lines 247-248.

  1. Claim

o  "The abundance of over-the-counter medications in Jordan makes it easy for patients to self-medicate for any disease at any time."
Issue The phrase "any disease at any time" is overly broad and not substantiated by the study.
Suggested Rephrase

  • "The availability of over-the-counter medications in Jordan facilitates self-medication for various conditions, including ophthalmic diseases."
  • - Thank you for this comment. We have now addressed this point, see lines 234-235.

General Recommendations

  • Use cautious language like "may," "suggests,"or "could indicate" when discussing findings that are not directly supported by data.
  • Avoid making causal claims unless the study design and analysis provide robust evidence.
  • Ensure all statements are directly supported by the data or references cited in the manuscript.

- Thank you for your valuable comments. We have now thoroughly revised our manuscript based on your recommendations and suggestions.

Comments on the Quality of English Language

Recommendations for improvement of the quality of the English language

  1. Proofreading Conduct a thorough review for grammatical errors, awkward phrasing, and redundancy.
  2. Professional Editing Consider using a professional editing service to ensure the manuscript adheres to the highest standards of academic English.
  3. Sentence Variety Vary sentence structure to avoid monotony and enhance engagement.
  4. Focus on Conciseness Remove unnecessary words or phrases to make the manuscript more succinct.

- Thank you for your valuable comments. We have now addressed these points.

Round 2

Reviewer 1 Report

Comments and Suggestions for Authors

Thanks for the revision!
The article has improved considerably.
The additional explanation needed is again for table 1. The age-groups: the first age group has been mentioned as 18-25 (7 yrs period), where other groups are having 4 years period. Pl. clarify! Or pl. explain at the foot of the table! 
pl. also explain why authors have selected a difference of 4 years, not 5 years. 

Author Response

Thank you for this comment. We have now justified this point in the footnote for Table 1 based on the reviewer comment.